# TextFormats: Simplifying the definition and parsing of text formats in bioinformatics

**Giorgio Gonnella**⬚*

Department of Bioinformatics, Institute of Microbiology and Genetics (IMG), University of Göttingen, Göttingen, Germany

\* giorgio.gonnella@uni-goettingen.de

## Abstract

Text formats are common in bioinformatics, as they allow for editing and filtering using standard tools, as well as, since text formats are often human readable, manual inspection and evaluation of the data. Bioinformatics is a rapidly evolving field, hence, new techniques, new software tools, new kinds of data often require the definition of new formats. Often new formats are not formally described in a standard or specification document. Although software libraries are available for accessing the most common formats, writing parsers for text formats, for which no library is currently available, is a very common though tedious task, utilized by many researchers in the field. This manuscript presents the open source software library and toolset *TextFormats* (available at https://github.com/ggonnella/textformats), which aims at simplifying the definition and parsing of text formats. Formats specifications are written in a simple data description format using an interactive wizard. Automatic generation of data examples and automatic testing of specifications allow for checking for correctness. Given the specification for a text format, *TextFormats* allows parsing and writing data in that format, using several programming languages (Nim, Python, C/C++) or the provided command line and graphical user interface tools. Although designed as a general purpose software, the main target application field, for the above mentioned reasons, is expected to be in bioinformatics: Thus, the specifications of several common existing bioinformatics formats are included.

## Introduction

Bioinformatics employs a multiplicity of data and file formats [1–3]. In many cases, these are text formats, or binary formats which can be easily converted to text representations for visualization and editing.

In a text format, information is represented by letters, numbers and symbols, each coded by a single byte or a small number of bytes using a general-purpose convention. Text representation codes are defined in standards, such as ASCII [4] and Unicode [5]. Conversely, binary representations use data sizes, order of the information and coding conventions, which are all specific to the type of data, to the application and often to technical details, such as the operating system and processor architecture [6].

**Data Availability Statement:** The TextFormats source code is available at the GitHub repository https://github.com/ggonnella/textformats.

**Funding:** Giorgio Gonnella has been supported by the DFG Grant GO 3192/1-1 "Automated

characterization of microbial genomes and metagenomes by collection and verification of association rules". The funders had no role in study design, data collection and analysis, decision to publish, or preparation of the manuscript.

**Competing interests:** The authors have declared that no competing interests exist.

Binary formats have some advantages: Since they are often similar to, or even directly reflect, the content of the working memory of the program, they are more efficient in terms of data access speed. Furthermore, binary formats require often less space than uncompressed text formats, since the information can be efficiently packed using representations tailored to the type of data.

Nevertheless, text formats remain very common and new formats are often defined in this form. Some features explain their popularity. First, the information in text formats can be accessed and often manually read or edited, without the need for the original software which produced the file. Text formats are accessible on different computer systems regardless of register size (e.g. 32 bit vs 64 bit) and byte order convention (little or big endianness); sometimes minor differences do exist, such as different newline conventions in different operating systems, but these are easily resolved, since they are often automatically handled by standard tools and functions. Finally, the data in text formats can often be examined, filtered and modified using a large number of standard command line tools (such as the Posix tools `sort`, `uniq`, `head`, `tail`, `cut`) or short scripts.

General purpose standards exist for representing information as text, such as XML, YAML and JSON. However, their adoption in bioinformatics is limited, likely because formats based on these standards are rather verbose and less human readable, due to their complex formatting and nested structure. Since they are not line-oriented, command line tools such as the one mentioned above, cannot be generally applied to these formats.

In recent years, community efforts have been made to define standard text formats for common types of data, such as GFA [7]. A goal of these is to avoid a further proliferation of formats. However, this is not easy to achieve in an open community of researchers. In the case of GFA, four variants currently exist (GFA1 [8], GFA1.1 [8], GFA2 [9], rGFA [10]) as a result of disagreements among researchers and the need to make the format particularly suitable to different applications. This case exemplifies the mechanisms by which new formats are often defined.

When a new software tool defines a new output format, the developer does not always provide a parser for the format, but often only a written documentation text. Formal grammars could be a solution to this problem by allowing the automatic generation of a parser with tools such as `yacc` or `bison`. However, they are challenging to write and rarely used in bioinformatics. Software libraries eventually become available for accessing new formats once they become popular. However, this process can take time, and parsers for less common formats are never or only partially implemented. Thus, whenever a researcher desires to programmatically access the data, he must write a parser based on the available specification or free text description. This often involves writing complex regular expressions, an error-prone and tedious task. The development of parsers is often repeated multiple times when switching languages, e.g., if a software project moves from rapid prototyping phase in Python to a more efficient implementation in C or C++.

Hereby, we present an open source free software project, named *TextFormats*, consisting of a software library and a collection of software tools. Its goal is to simplify the formal definition of new text formats, as well as provide easy and convenient access to the data represented in text formats, for which a parsing library does not yet exist. Given a format specification, *TextFormats* can be used for reading, validating and writing data in the format, from code in multiple programming and scripting languages (Nim, C, C++, Python, shell) as well as from the command line or using a graphical user interface. The library is versatile, allowing for sharing common sub-definitions among different formats, and provides a set of tools including an interactive format definition wizard, a specification format validator and an automatic example generator. Furthermore, examples applications (written in different programming languages) and specifications for common bioinformatics formats are included.

## *TextFormats*: Implementation and features

The core of *TextFormats* 1.0 is a software library implemented in the programming language Nim (v.1.6). It is accompanied by a suite of command line and graphical user interface tools, as well as modules for importing and using *TextFormats* in Python and in C or C++ programs.

*TextFormats* can be used for accessing information stored in a text format, provided that the format has been described in a specification written in *TFSL* (Text Formats Specification Language). A specification describes the representations of each single piece of information in the format, and expresses validation and transformation rules, if necessary. TFSL is a simple language for data description, described below.

Once a specification for a text format is available, *TextFormats* allows parsing of data in that format. Each piece of information in the text representation is thereby extracted, validated and transformed (if necessary) as described in the specification, and finally represented using an appropriate binary data type (e.g., numeric, string, array or dictionary). The opposite operation is also available, i.e., suitable data can be written in the format, using the representation described in the specification.

### The Text Formats Specification Language

The Text Formats Specification Language, briefly *TFSL*, is a declarative data format description language, developed as mean of describing a text format, in a concise and human readable manner. Typically a specification involves defining the format of each single piece of information in the representation, and combining simple data type descriptions into increasingly complex compound data types, until the entire data of a file or object has been described.

The valid syntax of a *TFSL* specification is described in the provided documentation, including the *TFSL* syntax manual, a *how to* manual with several examples, and a quick reference sheet. From here on, some of the main features of the language are summarized. Although the language is relatively simple, it is worth noting that the user does not necessarily have to learn the *TFSL* language, since a command line wizard tool `tf_genspec` can be used to generate interactively a *TFSL* file.

The information in a *TFSL* specification can be represented as a tree, where internal nodes have a string label (from a set of keys applicable in a given context) and the leaves of the tree contain scalar or compound data (strings, numeric values, boolean values, undefined values, lists or dictionaries). An example of a specification tree and the corresponding specification is given in Fig 1. The tree can be constructed programmatically, using a hierarchy of appropriate data structures, such as Python *dict* or Nim *table* objects, or can be written as a file in YAML 1.2 or JSON format.

The outermost level, under the tree root, defines a number of sections of the specification. Specifications usually define a number of datatypes, describing any piece of the information in the format and combined hierarchically in compound datatypes: These definitions are located in the section `datatypes` of the specification. The optional `testdata` section may contain examples of valid and invalid data in each of the defined datatypes, allowing for automatic testing of the specification (see next section).

Sometimes a definition requires one or multiple subordinate definitions, such as the format of elements of a list. In such cases, those definitions can be given inline or as a reference, to the name of another datatype, defined elsewhere. Thereby circular references are not allowed. Since a format often re-uses parts of other formats, it is possible to import definitions from a specification into another. Thereby, the `include` section allows to import single datatypes or entire specifications from one or multiple external files. Some components of imported definitions can be rewritten. In order to avoid naming conflicts, it is possible to use the `namespace`

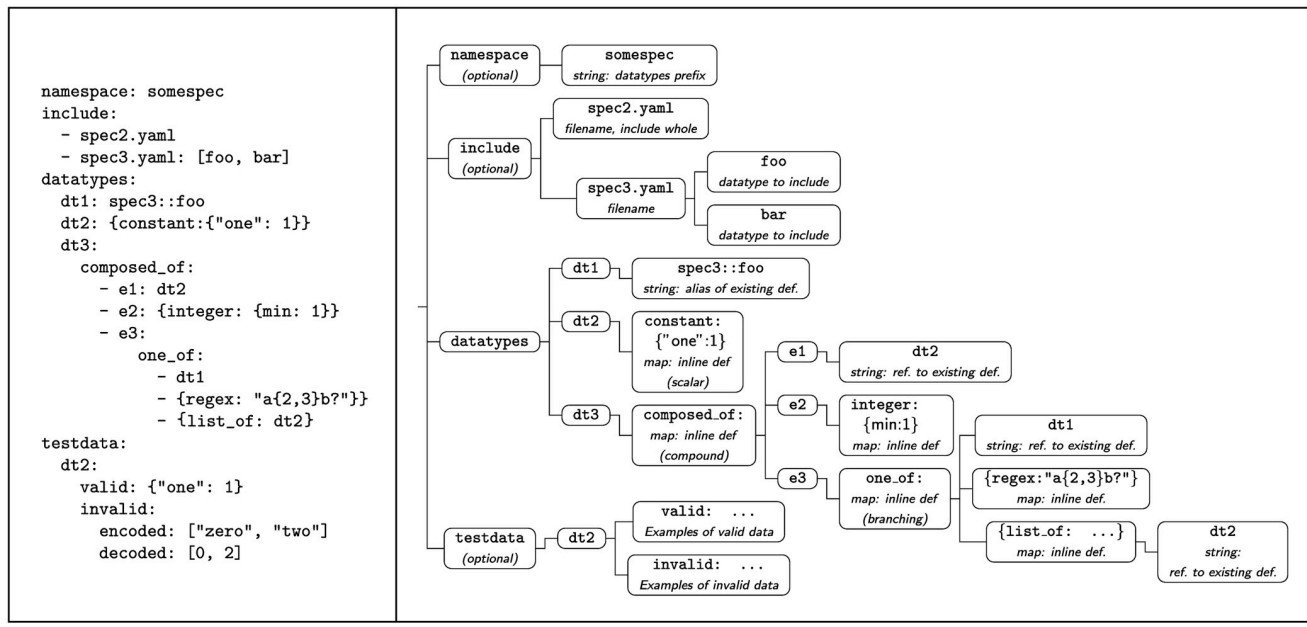

**Fig 1. View of a TFSL specification as a tree.** An example of *TextFormats* specification in YAML format (left) and the the information contained in the specification viewed as a tree (right).

section, to specify a namespace prefix used when datatypes are exported to another specification.

Multiple types of datatype definitions have been implemented, including scalar and compound values. A brief description of each kind of definition is provided in Table 1. Furthermore, definitions can contain different kinds of rules. Validation rules determine conditions which must be met by the represented data. Formatting rules define details of the text

**Table 1. Kinds of datatype definitions in the *Text Formats Specification Language*.**

| Structure | Group | Definition key | Description |
|---|---|---|---|
| Scalar | Discrete values | `constant` | only one value is valid |
| | | `values` | value must be the element of a set |
| | Regular expressions | `regex` | string matching a regular expression |
| | | `regexes` | string matching one of a set of regular expressions (optionally associated to different data transformation rules) |
| | Numeric intervals | `integer` | Signed base-10 integer, in a specified interval. |
| | | `unsigned_integer` | Unsigned base-2, -8, -10 or -16 integer, in a specified interval. |
| | | `float` | Floating point number in a specified open or closed interval. |
| Compound | Unordered sequences | `list_of` | list of elements, each with the same datatype or one of a set of datatypes, not depending on the element position in the list |
| | | `labeled_list` | list of elements, each associated to a string label (in a given set), defining the semantics and datatype of the element |
| | | `tagged_list` | list of elements, each associated to two string labels, defining, respectively, semantics and a datatype of the element |
| | Ordered sequences | `composed_of` | ordered sets of elements, each with a possibly different datatype |
| Scalar/ compound | Alternatives | `one_of` | multiple alternative valid representations |

representation, such as constant prefixes/suffixes or field delimiters. Transformation rules allow to map text representations to the value they represent: e.g., the symbol `D` to the string *deletion*, the roman value *III* to the integer 3, or strings such as `T` and `F` to boolean values.

## Tools for the validation and testing of specifications

Once a format has been defined using *TFSL*, it is possible to check if the definitions are correct and indeed define the format as intended. Two tools are available for this task: `tf_spec info` and `tf_cerberus.py`. The command line tool `tf_spec info` checks that a specification is correct (and outputs a list of datatypes in that case). In case a specification has a mistake, an extensive help message is displayed, with a summary of the valid syntax in the context of the error. In some cases, however (for example when the structure of the YAML/JSON file is invalid), the tool `tf_cerberus.py` (based on the Python validation library *Cerberus*) is more useful for localizing the error.

Even if the syntax of a specification is correct, it is still possible that it does not accurately describe the target format. Two tools can be combined to avoid this. First, examples of valid and invalid data for each of the datatypes defined in a specification can be added to the specification file (or to a separate test file). By using the command line tool `tf_spec test` (or the corresponding API functions), these validity expectations can be automatically tested. Furthermore, examples of valid and invalid data (formatted as test data) for each of the datatypes can be automatically generated using the command line tool `tf_spec gentest`. The user can inspect the generated examples, to check if they reflect the intended format representation. Those examples can also be modified to generate more expected valid and invalid data to include in the specification tests.

## Compilation of TFSL specifications

When a specification is provided to the *TextFormats* library, first the YAML or JSON structure is parsed, then the *TFSL* syntax is validated. Any included specification is then parsed; This operation is done recursively, as included specification may include other files themselves.

Next, all datatype definitions are parsed from the specification being processed, as well as included specification. Datatype definitions can refer to other definitions, on which they depend: e.g., a list depends on the definitions of its elements, which can be given as a reference in the specification. Thus, a directed acyclic graph of the dependencies of the datatype definitions is created. The graph is employed to check for the absence of circular dependencies, using a depth-first cycle detection algorithm, and to solve all references. A hash table of datatype names to datatype definitions is then created.

For each of the datatype definitions, a regular expression is computed and compiled using the Nim `regex` library. Among the available Nim libraries for regular expressions, this library was chosen since it offered better performance, pre-compilation of the regular expressions and better handling of capturing groups (S2 Appendix). The regular expressions are employed for validation and parsing of compound data in the text representation, except in some cases where other strategies are used, such as splitting by a fixed exclusive text delimiter.

The compilation steps summarized above are done, by default, *just-in-time* when the YAML or JSON file is loaded. Examples of running time of the compilation phase are given in Table 2. Although compilation is very fast for all provided example specifications, the overhead introduced by the compilation steps may be reduced, by saving the compiled specification to file. This operation is performed using the command line tool `tf_spec compile` or equivalent API functions. Pre-compiled specifications can be used instead of a YAML or JSON specification in all *TextFormats* tools and functions. However, since parsing the compiled

**Table 2. Time for compilation of TFSL specifications from YAML files and loading of pre-compiled specifications.**

| Format | Compile only | Compile and save pre-compiled | Load pre-compiled |
|---|---|---|---|
| Accessions | 0.02 s | 0.02 s | **0.02 s** |
| NCBI ID | 0.03 s | 0.03 s | **0.02 s** |
| Fasta | **< 0.01 s** | 0.01 s | 0.01 s |
| FastQ | 0.01 s | 0.01 s | **< 0.01 s** |
| SAM | **0.25 s** | 0.21 s | 0.27 s |
| EGC | 0.23 s | 0.24 s | **0.08 s** |
| GFA1 | **0.26 s** | 0.31 s | 0.45 s |
| GFA2 | 1.28 s | 1.40 s | **1.19 s** |
| GFF3 | 1.79 s | 1.79 s | **1.25 s** |

bold font indicates the fastest time for obtaining the specification: loading a pre-compiled specification or compiling the YAML specification.;

The operations were performed using the *TextFormats* command line tool `tf_spec`, with the sub-commands `compile` (compile and save to file) and `info` (compile YAML or load pre-compiled).

The reported times are the average over 3 runs of the real time measured by GNU *time*, on a Linux OpenSuse 15.3 workstation (CPU i5–4570 3.2 Ghz, RAM 16 Gb), using *TextFormats* 1.2.2.

specification from file and reconstructing the objects in memory also requires time, pre-compiling is only meaningful for particularly complex specifications, when these are loaded multiple times (e.g., when decoding multiple strings with the *TextFormats* command line tools).

Table 2 shows the time necessary for parsing and compiling the YAML specification and output a list of datatypes, save the compiled specification to file and listing the datatypes from the pre-compiled specification.

## Operations on text formats

Provided a *TFSL* specification of a text format, *TextFormats* implements a number of operations for handling data in that format. Table 3 summarizes the available operations.

The core operations are *decoding*, i.e., converting the text representation into the data it represents, and *encoding*, i.e., writing a text representation, given some suitable data. A *validation* operation is also available, which can be applied to the text representation to check if it follows the specification, or to the decoded data to check if they can be represented in the specified format. In some cases this operation can be done without requiring full decoding or encoding, e.g., simply applying the regular expression for the given datatype.

The input for the decoding operation can be a string in the text format or a file. When a file is given as input, the decoder must know the definition scope, i.e., to which part of the file the datatype definition shall be applied. In particular, this can be: a single line (*line* scope); a fixed number of lines of the file (*unit* scope); all following lines which were not yet processed lines to which it applies (*section* scope); the entire file (*file* scope). Definition with section and file scope are useful for validating the structure of the entire data: e.g., in a SAM format file [11], there must be a header section followed by an alignment section. This validation is not possible if single lines are parsed independently. However, processing the parsing results all at once would require a large amount of memory e.g., if a large file is parsed. In such cases, it is possible to instruct the decoding function to process only one line at a time (or one element of a compound definition at a time), but still validate the entire data. For example, for a SAM file,

**Table 3. Operations implemented by *TextFormats* and corresponding API functions and CLI commands.**

| Input | Operation | Interface | Function/Command |
|---|---|---|---|
| Specification | Compile *TFSL* specification | Nim | `filename.compile_specification()` |
| | | C | `tf_compile_specification()` |
| | | Python | `Specification.compile()` |
| | | CLI | `tf_spec compile` |
| | Load *TFSL*/compiled specification | Nim | `filename.specification_from_file()` |
| | | C | `tf_specification_from_file()` |
| | | Python | `Specification()` |
| | | CLI | `-s/--spec` option of all commands |
| | Run tests | Nim | `run_specification_testfile()` |
| | | C | `tf_run_specification_testfile()` |
| | | Python | `Specification.test()` |
| | | CLI | `tf_spec test` |
| Text representation | Validate | Nim | `string.is_valid()` |
| | | C | `tf_is_valid_encoded()` |
| | | Python | `DatatypeDefinition.is_valid_encoded()` |
| | | CLI | `tf_validate encoded` |
| | Decode (input: string) | Nim | `string.decode()` |
| | | C | `tf_decode()` |
| | | Python | `DatatypeDefinition.decode()` |
| | | CLI | `tf_decode string` |
| | Decode (input: file) | Nim | `filename.decode_file()` |
| | | C | `tf_decode_file()` |
| | | Python | `DatatypeDefinition.decode_file()` |
| | | CLI | `tf_decode file` |
| Data | Check if suitable for representation | Nim | `jsonnode.is_valid()` |
| | | C | `tf_is_valid_decoded()` |
| | | Python | `DatatypeDefinition.is_valid_decoded()` |
| | | CLI | `tf_validate decoded` |
| | Encode into text representation | Nim | `jsonnode.encode()` |
| | | C | `tf_encode()` |
| | | Python | `DatatypeDefinition.encode()` |
| | | CLI | `tf_encode json` |

the decoder would still validate the correctness of the global structure of the file, but it would process only one header line or alignment at a time.

## Supported programming languages

The *TextFormats* library has been implemented in the Nim programming language (version 1.4.8). This language offers a number of advantages over alternatives (it is compiled, but faster to code than C/C++) and has recently aroused interest [12] and some limited adoption [13, 14] in the bioinformatics community. A reason why this language was chosen for this project is the ease of interfacing Nim code to other programming languages. Thus, besides using the library in Nim, also C/C++, Python and command line scripts are supported (see S1 Appendix for code examples in Nim, Python, C and Bash). The following sections briefly describe the implementation challenges, design choices and peculiarities of these interfaces.

**The C API to *TextFormats*.** For using *TextFormats* in C and C++ the library and the C API modules are compiled and linked to the Nim runtime library, and the resulting header file is included into the C or C++ program. The C/C++ API modules functionality is documented in a manual, as well as in a quick reference sheet. The core module is a wrapper to the Nim API functions for use in C (implemented using the *exportc* Nim pragma). However, additional module had to be implemented, to cope with the differences between C and Nim.

Nim is a statically typed language, like C. However, the datatype of data obtained by decoding a text representation is not know at compile time. In Nim this problem has been solved by employing a *variant type* from the standard library (`JsonNode`), which can represent different kinds of data and provides multiple functions for accessing and modifying the data. In order to use the same strategy in C and to provide a consistent interface, a wrapper to the Nim `json` library was developed and included in the *TextFormats* C API.

A further challenge is represented by exceptions, since in C there is no exception handling, equivalent to that in Nim. Thus a mechanism similar to the *errno* of the C standard library has been adopted. In particular, if a *TextFormats* function results in an exception, a variable describing the error is set. The user can decide to handle the exception or print an error message and quit the program. Alternatively, to avoid code redundancy, it is also possible to specify, with a single line of code, that all errors must result in printing the error message and quitting.

**The Python API to *TextFormats*.** Python is a very popular choice for developing bioinformatics pipelines. It is easy to import Nim code into Python using the Nim library `NimPy` v.0.1 (available at https://github.com/yglukhov/nimpy), and the Python library `nimporter` (available at https://github.com/Pebaz/nimporter) v.1.0.4.

However, a simple wrapper to the Nim functions results in a functional but inelegant interface. Therefore a Python API module has been developed on top of it, which defines classes representing *TextFormats* specifications and datatypes. The module allows adoption of a more idiomatic coding style, with greater reflection of the dynamic typing and object orientation of Python. A manual and a quick reference sheet describe the use of this API.

**The command line interface to *TextFormats*.** Bioinformatics analyses often involve executing multiple programs, which can be combined using command line scripts. To enable the use of *TextFormats* in this context, a suite of command line interface (CLI) tools has been developed. Their usage is documented in a manual, in `man` pages for each of the tools, and in a quick reference sheet. The tools support the use of standard input and output, in order to facilitate their inclusion in pipes.

The decode, encode and validate operations of *TextFormats* are provided, respectively, by the `tf_decode`, `tf_encode` and `tf_validate` tools. The `tf_spec` provides operations on specifications, such as analysis of their content, testing, automatic generation of example data, and pre-compilation of *TFSL*.

## Results

### Case study 1: Parsing a complex format

In order to test the *TextFormats* library on real world data, we implemented a `SAM` format [11] *TFSL* specification, based on the format specification document [15]. We implemented several versions of a program for counting the alignments by target sequence, by read group, by flag value, and the occurrences of each optional tag found in the file.

First, we compared the performance of *TextFormats* when using it from Nim, or from other languages. Thus we implemented the parser, based on *TextFormats*, in Nim, Python and C. Furthermore, we implemented the same functionality without *TextFormats* and used

**Table 4. Running times of equivalent programs based on *TextFormats* or other libraries, implemented in Nim, Python, and C.**

|  | N. input lines | Library | Nim | Python | (vs Nim) | C | (vs Nim) |
|---|---|---|---|---|---|---|---|
| (SAM) Case study 1 | 100 000 | *TextFormats* | 5.45 s | 5.70 s | (+ 4.6%) | 5.61 s | (+ 2.9%) |
|  | 500 000 | *TextFormats* | 26.89 s | 28.14 s | (+ 4.6%) | 27.76 s | (+ 3.2%) |
|  | 1 000 000 | *TextFormats* | 53.70 s | 55.91 s | (+ 4.1%) | 55.46 s | (+ 3.3%) |
|  | 100 000 | htslib | 0.35 s | 2.44 s |  | 0.09 s |  |
|  | 500 000 | htslib | 1.66 s | 12.16 s |  | 0.42 s |  |
|  | 1 000 000 | htslib | 3.37 s | 24.33 s |  | 0.83 s |  |
| (EGC) Case study 3 | 100 000 | *TextFormats* | 5.38 s | 5.87 s | (+ 9.1%) | 5.31 s | (- 1.3%) |
|  | 500 000 | *TextFormats* | 25.78 s | 28.33 s | (+ 9.9%) | 25.40 s | (- 1.4%) |
|  | 1 000 000 | *TextFormats* | 52.74 s | 55.70 s | (+ 5.6%) | 51.62 s | (- 2.3%) |
|  | 100 000 | *ad hoc* | n.a. | 2.19 s |  | n.a. |  |
|  | 500 000 | *ad hoc* | n.a. | 11.37 s |  | n.a. |  |
|  | 1 000 000 | *ad hoc* | n.a. | 22.69 s |  | n.a. |  |
| (GFA2) (Case study 4) | 363 613 | *TextFormats* | 93.55 s | 96.83 s | (+ 3.5%) | n.a. |  |
|  | 363 613 | GfaPy | n.a. | 191.83 s |  | n.a. |  |

(SAM) Case study 1: program for collecting information from a SAM file, based on the *TextFormats* or the *htslib* library;

(EGC) Case study 3: program for parsing the EGC format (defined in the text) writing the information to JSON and then back to EGC, based on the *TextFormats* library, or as a *ad hoc* Python parser;

(GFA2) Case study 4: Python program for validating a GFA2 file and collecting basic statistics on the file, based on *TextFormats* library or on the *GfaPy* library;

(vs Nim) Running time difference of the Python or C version (when implemented) of the *TextFormats*-based programs to the Nim version;

The reported times are the average over 3 runs of the real time measured by GNU *time*, on a Linux OpenSuse 15.3 workstation (CPU i5–4570 3.2 Ghz, RAM 16 Gb), using *TextFormats* 1.2.2.

instead the state-of-the art library *htslib* [16] v.1.13. Also in this case, we compared the native implementation in C, with the use of the Python wrapper *Pysam* [17] v.0.17.0 and of the Nim wrapper *hts-nim* v.0.3.18 [13].

As test data, we used a SAM file from the 1000 Genomes Project [18], the Mosaik alignment of the 454 sequencing of sample NA06984 (file NA06984.454.MOSAIK.SRP000033.2009_11. bam obtained from http://ftp.1000genomes.ebi.ac.uk/vol1/ftp/pilot_data/data/NA06984/ alignment/ and converted to SAM using samtools [11]).

We measured the running time of each of the implementations as real time measured by GNU *time* [19] v.1.9 (average of 3 runs, run on a Linux workstation with CPU Intel i5–4570 3.20GHz, 16 Gb RAM, Linux OpenSuse 15.3). The results are summarized in Table 4. The same counts were output by each version of the program, based on *TextFormats* or *htslib*, in Nim, Python and C.

## Case study 2: Parsing sequence identifiers

The sequences contained in sequence databases are identified by accessions, which remain stable when corrections or new versions of the same sequence or sequence annotation are published. Accessions are strings consisting of sequences of letters and numbers. The valid formats of accessions are described in the documentation of the databases. Besides a number identifying the entry, accessions often include more information, such as the section of the database, or the type of molecule or annotation.

In contrast to file formats such as SAM (discussed in Case study 1), there is no available parser or validator for accession strings. Thus we implemented the *TFSL* specifications spec/

**Table 5. Accession identifiers of NCBI, DDBJ, ENA/EBI and UniProt sequence databases defined in the `spec/accessions.yaml` *TextFormats* specification.**

| Database | Data coded in accession |
|---|---|
| INSD read archives (SRA, DRA, ERA) | Institution (NCBI, DDBJ, ENA/EBI), Type of data (study, run, sample, experiment, analysis), Entry |
| UniProtKB | Database name, Entry |
| Trace Archive | Database name, Entry |
| INSD assembled sequence (Nucleotide, Protein, Bulk, MGA) | Database name, Entry |
| INSD metadata (BioProject, BioSample) | Institution (NCBI, DDBJ, ENA/EBI), Type of Record (BioProject, BioSample), Entry |
| RefSeq | Type of molecule (Genomic, RNA, protein), Type of assembly (reference, alternate), Type of annotation (curated, predicted model), Entry |
| Ensembl | Species, Feature type (exon, protein, gene, transcript etc), Entry |

The definitions on which the specification is based were obtained from the following documentation pages: https://www.ncbi.nlm.nih.gov/Sequin/acc.html, https://www.ddbj.nig.ac.jp/acc_def-e.html, https://www.ddbj.nig.ac.jp/prefix-e.html#dra, https://www.ncbi.nlm.nih.gov/books/NBK21091/table/ch18.T.refseq_accession_numbers_and_mole/, https://www.uniprot.org/help/accession_numbers and https://www.ensembl.org/info/genome/stable_ids/prefixes.html.

`accessions.yaml`, describing the format of the accessions of multiple sequence databases (Table 5), and `spec/ncbi_id.yaml`, describing the sequence identifiers used by NCBI for sequences in Fasta format (Table 6). They allow for effortless validation of the identifiers and access to the information contained therein, from the command line or any of the supported programming languages (Nim, Python, C, C++).

## Case study 3: Defining a new format

New text formats are often defined to represent kinds of data for which no existing suitable format yet exists. One of the goals of the *TextFormats* library is to simplify the definition of new formats in those circumstances. To simulate this kind of application, we defined a new file format and implemented its specification in *TFSL* (`egc.yaml`). The format, called EGC (*expected genomic content*) has the goal of representing a set of rules, describing the expected content of a microbial genome, under a given condition, such as phenotype, lifestyle, or membership in a taxonomic group.

The general structure of the format was organized on the example of the GFA format [9]. Each line not starting with a comment symbol (#) is a record, containing multiple fields, separated by tabulator characters. Tabulators or newline characters never occur in these fields. The first field is a single letter determining the record type. The number and semantics of the following positional fields are determined by the record type. The positional fields cannot be empty and a point (.) is used to represent missing information in a field (whenever allowed).

Four types of record lines have been have defined in EGC: records of type A define attributes which can be measured in a genome sequence or annotation, such as sequence statistics or feature counts; records of type T (taxon) and P (phenotype group) define measurement subjects, i.e., groups of organisms for which an expected value of the attributes can be defined; finally records of type E define the expectation, i.e. the association of a subject to expected values of an attribute.

**Table 6. Fasta sequence identifiers used by NCBI, defined in the `spec/ncbi_id.yaml` *TextFormats* specification.**

| Type of sequence | Accession prefix | Example |
|---|---|---|
| NCBI RefSeq database | `ref` | `ref\|NM_010450.1` |
| NCBI GenBank database | `gb` | `gb\|M73307\|AGMA13GT` |
| NCBI GenBank (third-party annotation) | `tpg` | `tpg\|BK003456\|` |
| EMBL sequence database | `emb` | `emb\|CAM43271.1\|` |
| EMBL sequence (third-party annotation) | `tpe` | `tpe\|BN000123\|` |
| DDBJ sequence database | `dbj` | `dbj\|BAC85684.1` |
| DDBJ sequence (third-party annotation) | `tpd` | `tpd\|FAA00017\|` |
| SWISS-Prot database | `sp` | `sp\|P01013\|OVAX_CHICK` |
| TrEMBL database | `tr` | `tr\|Q90RT2\|Q90RT2_9HIV1` |
| PIR database | `pir` | `pir\|\|G36364` |
| PDB database | `pdb` | `pdb\|1I4L\|D` |
| PRF database | `prf` | `prf\|\|0806162C` |
| patent sequence | `pat` | `pat\|US\|RE33188\|1` |
| pre-grant patent sequence | `pgp` | `pgp\|EP\|0238993\|7` |
| general database reference | `gnl` | `gnl\|taxon\|9606` |
| local sequence | `lcl` | `lcl\|hnm271` |
| GenInfo backbone sequence ID | `bbs` | `bbs\|316342` |
| GenInfo backbone molecule type | `bbm` | `bbm\|464147` |
| GenInfo import ID | `gim` | `gim\|442187` |
| GenInfo integrated database | `gi` | `gi\|21434723` |
| NCBI internal, genome pipeline | `gpp` | `gpp\|GPC_123456789` |
| NCBI internal, named annotation track | `nat` | `nat\|AT_123456789.1\|` |

The format of each type of identifier is described in the documentation of the NCBI Toolkit, at https://ncbi.github.io/cxx-toolkit/pages/ch_demo#ch_demo.id1_fetch.html_ref_fasta.

We developed a parser for the EGC format using *TextFormats*. In order to quantify the possible overhead when implementing *TextFormats*-based programs in different programming languages, we implemented the program in Nim, C and Python.

Furthermore, to compare the use of *TextFormats* with existing solution, we developed a Python parser for the format, which does not rely on *TextFormats*. We could not find any suitable Python library for easily creating such a parser. For example, GfaPy [20], which allows to read GFA2 files, can be extended to new datatypes and custom line types. However, this functionality is meant for adding further structured information to the graph, and does not fit the need to implement a format not aimed at representing a graph (e.g., the standar GFA2 record types cannot be overwritten). Thus, we created an *ad hoc* EGC parser in Python from scratch (`egc_ad_hoc.py`).

The results obtained with the *ad hoc* parser were identical to those obtained using the programs based on *TextFormats*. We compared the performance of the different implementations on example files, containing a variable number of lines. The results are reported in Table 4.

## Case study 4: Development of a Python software tool

Using the Python API of *TextFormats*, it is possible to rapidly develop software tools addressing complex formats, such as GFA2. To demonstrate this, we created a Python script `gfa2_info.py` based on the library, which collects basic statistics and summarizes the

contents of a GFA2 file. To compare this solution with the state-of-the-art and verify the results, we developed a tool with the same functionality using another software library. Among the existing GFA libraries, only GfaPy [20] allows parsing of a GFA2 file using Python. Thus we developed a script (named `gfa2_info_gfapy_based.py`) based on it.

In the current implementation of *TextFormats*, constraints which involve non-consecutive elements cannot be directly specified in the specification, but must be implemented in the calling code. In GFA2, all record identifiers must be unique, references to segments in other lines must be identifiers of segments defined elsewhere in the same file, and the coordinates in edges must be in the range of the length of the sequences to which they refer and correctly use the final coordinate marker. To exemplify the implementation of such constraint validations when using *TextFormats* and ensure a fair comparison with GfaPy, we developed a module `gfa2_cross_validator.py`. The module verified the constraints when running the *TextFormats* version of `gfa2_info.py`: it was able to correctly identify and report multiple issues in an example GFA2 file.

Furthermore, in order to quantify the possible overhead when implementing the programs based on *TextFormats* in Python, we implemented an equivalent *TextFormats*-based program also in Nim, including a Nim implementation of the cross validator module.

We tested the Gfapy- and *TextFormats*-based programs on large real data, using the the Minigraph [10] pre-built human genome pangenome graph `GRCh38-0.1-14.gfa.gz` (downloaded from ftp.dfci.harvard.edu/pub/hli) converted to GFA2 by GfaPy. The file consists of about 363 thousand lines. All programs produced the same results. The running times are reported in Table 4.

## Case study 5: Data format standardization

Sometimes data is available in a custom format and requires conversion into a standard format, in order to be processed with existing software tools. To test the suitability of *TextFormats* for this kind of task, we created a file containing the annotation of a gene in a custom tabular format.

A *TextFormats* specification was then written to read the custom file (`ftab.yaml`). We then created a Python script (`ftab_to_gff3.py`), which parses the custom tabular format using *TextFormats* and re-organizes the information, so that it can be written in GFF3 format, using the provided GFF3 *TextFormats* specification. The resulting file was correctly validated by the online GFF3 validation tool http://genometools.org/cgi-bin/gff3validator.cgi of the GenomeTools suite [21].

## Case study 6: Repairing an invalid file

Sometimes, due to some issues, a software tool outputs a file, which is invalid according to its format specification. When it is not possible to fix the software tool, the output file must be edited and corrected, so that it can be further processed with other tools, which assume a correct format. In the case of a complex format the correction can be very cumbersome, since it is necessary to edit the invalid formatted pieces of information but existing library often interrupt parsing due to the format error, and thus they can not be employed to edit the file content and fix the issue.

For example, when extracting a sub-graph from a large GFA1 file, Bandage v.0.8.1 [22] outputs an invalid GFA1 file, which could not be loaded in standard-compliant GFA tools, such as GfaPy [20]. Using a *TextFormats*-based Python script, the invalid file was further investigated (`gfa1_show_invalid_lines.py`). This showed that the invalid tag type code 'z' was included in some tags (instead of the correct type code 'Z' for string types), and segment

lines without sequences did not include the necessary * symbol instead of the sequence. An example of GFA1 file which causes such an issue when a subgraph is extracted is included in the *TextFormats* package (`complete_graph.gfa`).

Here we show, how it is possible to use *TextFormats* for solving this problem. In particular, *TextFormats* allows import of an existing specification and changes to some parts of it. Thus, we created a new specification which describes the format of the corrupted file (`invalid_gfa.yaml`). In it, the original GFA1 specification was imported, the definition of tags was modified to include the incorrect tag code, and the definition of sequences was modified, so to accept empty strings instead of the * symbol.

We then created a short Python script based on *TextFormats* (`gfa1_fix.py`) which parsed the output of Bandage using the `invalid_gfa.yaml` specification and output the graph using the GFA1 specification. The resulting file was valid GFA1, which could be correctly parsed by GfaPy.

## Discussion and conclusion

*TextFormats* is a software library and toolset which aims at providing an easy system for the definition and access to text formats, which are very common in Bioinformatics. In particular, it provides a rapid prototyping solution to the tedious task of parsing formats for which a parsing library is not yet available. We tested the software by providing definitions of complex formats such as SAM (Case study 1). We compared the resulting SAM parser to the state-of-the art parsers based on the HTSlib library [16]. HTSlib resulted in much more efficient parsing and provides additional functionality, compared to the *TextFormats*-based application. However, the difference in the efforts required for implementing such a library is apparent when comparing the number of codes: HTSlib (as of version 1.13) consists of 84000 lines of code (and, of course, offers additional functionality). The SAM specification in the *Text Formats Specification Language* consists of a mere 132 lines. In another example, Case study 4, we implemented Python scripts for collecting statistics from GFA2 files. The script based on *TextFormats* and a TFSL specification for GFA2 (224 lines) and a Python cross-validation module (127 lines) had a better performance than a script based on the Python library GfaPy which (as of version 1.2.3) consists of over 10000 lines of code. Thus, we think that *TextFormats* represents a useful tool, a tradeoff between computational performance and development effort, when implementing bioinformatics scripts and pipelines, in which file formats must be accessed for which no software is yet available.

In many cases, bioinformatics formats are only defined in text documents. This is for example the case for accession numbers of sequence databases (Case study 2). *TextFormats* does not require the user to write the formal grammar for describing a format, a task which can be challenging and is rarely used in bioinformatics. It relies on a simpler, human readable, definition language *TFSL*. This hopefully will encourage authors of tools and databases to provide a specification to their data formats, instead of a mere description text. Such a specification could directly be used for working with the data in the format.

In Case study 3, we made an example of design and definition of a format from scratch, using *TextFormats* and compared this to the development of an *ad hoc* Python parser. While the *ad hoc* parser was faster in handling an example input file, the development effort was also much higher. *TextFormats* specification consisted of a 150 lines YAML file. The *ad hoc* parser code is much more complex and difficult to maintain: it consists of about 700 lines of Python code, for a total of 73 methods, aimed at parsing and writing all elements of the defined format. It necessarily mixes the format definition with code for parsing and writing data based on those definition. In contrast, using *TextFormats* the structure of the format is immediately

apparent from the format specification file. Thus it is very easy to change any element of the format, and even the whole structure of the file, which is very useful during the development of a new format. Also, *TextFormats* provides further functionality, such as testing and automatic examples generation from the format specification.

Also when adopted for reading or writing existing formats, *TextFormats* can be useful. In Case study 5, for example, we demonstrate the use of *TextFormats* for converting annotation data in a table to the standard GGF3 format. In another example (Case study 6), we demonstrated the correction of an invalid GFA1 file output by another tool, which was rejected by standard-compliant parsers. The *TextFormats* specification for GFA1 could be used for identifying invalid elements of the file. Without *TextFormats*, correcting these elements requires to correctly fetch them among the rest of the file content. For a complex format such as GFA1, performing this operation correctly requires parsing at least the relevant parts of the format. Thus, it would require to either write a parser from scratch or patch the source code of an existing parsing library for the format. In *TextFormats* the file correction much easier, as it allows overwriting definitions of imported specifications. Thus, a patched specification for GFA1 was easily constructed just overwriting the parts of the format defining the invalid elements (15 lines of TFSL specification).

Although we think that *TextFormats* can be very useful in applied bioinformatics, it is also has some limitations, which could be addressed in future versions of the software. First, its lower performance compared to ad hoc format parsers is partly inherent in the dynamic nature of the software, as *TextFormats* must employ flexible data structures for the representation of data, whose type is not known when compiling the library. In this context, an interesting feature of Nim, not used in the current implementation, is the ability to execute a subset of the language at compile time; A growing number of Nim libraries support this feature. It is conceivable to exploit this functionality by giving the Nim compiler further information about the types of data to be represented, given a *TFSL* specification. This would allow it to create versions of the software addressed at single formats only, but with higher performance.

Although *TextFormats* is written in Nim, a programming language which is rather unusual for bioinformatics software, the user of the library does not need to employ the same language. Instead, API for Python and C are provided. A major goal of the library is rapid development, and Python is a very popular rapid development language in the bioinformatics community. Thus, it is foreseeable that most user will employ the library through the Python API. An example of development of Python tools using *TextFormats* is given in Case study 4. It is worth noting that installing the Python library is very easy: The documentation includes details of the procedure, which in many cases do not even require a Nim compiler, but just to run the command *pip install textformats*.

It could be argued that a package mainly intended to be used from Python should be implemented in Python itself. To analyse the overhead represented by the use of the library in a different language than the implementation language Nim, we implemented equivalent *TextFormats*-based programs (Case study 1 and Case study 4) in Nim, Python and C. Since Nim is compiled to C, the overhead of using C instead of Nim itself is very limited: for the program described in Case study 1, it was 2.9% to 3.3% (Table 4), while the program described in Case study 3 runs slightly faster when written in C (1.3% to 2.3% faster). The overhead in Python was measured comparing the running times to Nim implementations of the programs described in Case study 1, Case study 3 and Case study 4. It was higher than in C, with values ranging from 3.5% to 9.9%. The additional time is required for the initialization of the Python interpreter (which would be required also if the library would be implemented in Python) and for the data exchange, which requires Python object initializations handled under the hood by the *Nimpy* library (for example for strings, which are in Python stored as immutable objects).

However, when using a compiled language, such as Nim, instead of Python, for developing Python libraries, the higher performance of compiled code compensates this additional time. For this reason several popular Python packages in science, such as Numpy and Scipy, are implemented as C extensions. Nim is compiled to C, and its Python interface is based on the same Python C extension API used by those packages.

One of the central features is the generation of regular expressions for the datatypes defined in the specification, from the description of the datatypes in *TFSL*. These regular expressions, in most cases, are used to parse the input and capture its components. As a consequence, a limitation of the library is that the formats that can be specified must be, in general, regular languages [23]. Another parsing strategy, not based on regular expressions, would be required to overcome this limitation. Fortunately, most bioinformatics text formats are regular languages. Still, some formats allow any degree of nesting of elements, e.g. the Newick format for phylogenetic trees [24], and thus cannot be currently described in the current version of *TFSL*. There is an exception to this limitation: JSON, including any level of recursion, can be embedded in any format supported by *TextFormats*. This is achieved by delegating the parsing of the embedded JSON to the Nim JSON library; this functionality could also be extended by interfacing additional external libraries.

A further current limitation of *TextFormats* is in the validation of data, whenever the comparison of non-adjacent pieces of information is necessary. For example, in a format representing a graph (e.g. GFA [8]), it is not possible to model in the specification the constraints that all edges must refer to valid nodes, since the nodes are defined elsewhere in the file. Currently, such additional validations can be implemented as an additional layer on the data parsed by *TextFormats*, as exemplified for the GFA2 format in Case study 4. In future versions of *TextFormats*, this validation layer could be generalized and integrated in the library. This will require a system for addressing each single part of a format definition and a temporary storage of information which must be used as comparison reference (e.g. sets valid of identifiers).

To conclude, we think that *TextFormats*, alongside current alternatives (such as writing parser scripts) and despite some limitations described above, is an useful and powerful system for rapidly supporting access to information in new bioinformatics text formats, as well as for the definition of new formats, by providing a simple but effective format definition language.

## Supporting information

**S1 Appendix. Example code based on *TextFormats*.** Examples of Python, Nim, Bash and C code using the *TextFormats* library for parsing a text format.
(PDF)

**S2 Appendix. Comparison of the available regular expression libraries for Nim.** Comparisons of the performance and features of the currently available regular expression libraries for the Nim programming language: re, nre, regex and nregex.
(PDF)

## Acknowledgments

Many thanks to Burkhard Morgenstern (Department of Bioinformatics, University of Göttingen), Marco Matthies and Stefan Kurtz (Center for Bioinformatics, University of Hamburg) for helpful discussions; to Serena Lam (Department of Bioinformatics, University of Göttingen) for language style suggestions and grammar corrections.

## Author Contributions

**Conceptualization:** Giorgio Gonnella.

**Funding acquisition:** Giorgio Gonnella.

**Investigation:** Giorgio Gonnella.

**Project administration:** Giorgio Gonnella.

**Software:** Giorgio Gonnella.

**Supervision:** Giorgio Gonnella.

**Writing – original draft:** Giorgio Gonnella.

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
