## [Decision Letter · Decision Letter 0]

17 Jan 2022

PONE-D-21-34058TextFormats: simplifying the definition and parsing of text formats in bioinformaticsPLOS ONE

Dear Dr. Gonnella,

Thank you for submitting your manuscript to PLOS ONE. After careful consideration, we feel that it has merit but does not fully meet PLOS ONE’s publication criteria as it currently stands. Therefore, we invite you to submit a revised version of the manuscript that addresses the points raised during the review process. In particular, in the revised manuscript (should you decide to revise), I would expect a comparative analysis as indicated by the reviewer. Adding another use case is also highly recommended.

We look forward to receiving your revised manuscript.

Kind regards,

M. Sohel Rahman, Ph.D.

Academic Editor

PLOS ONE

Journal Requirements:

“This work has been supported by the DFG Grant GO 3192/1-1 “Automated characterization of microbial genomes and metagenomes by collection and verification of association rules”.”

“Giorgio Gonnella has been supported by the DFG Grant GO 3192/1-1 ``Automated

characterization of microbial genomes and metagenomes by collection and

verification of association rules''.

Reviewers' comments:

Reviewer's Responses to Questions

**Comments to the Author**

1. Is the manuscript technically sound, and do the data support the conclusions?

Reviewer #1: Partly

2. Has the statistical analysis been performed appropriately and rigorously? 

Reviewer #1: N/A

3. Have the authors made all data underlying the findings in their manuscript fully available?

Reviewer #1: Yes

4. Is the manuscript presented in an intelligible fashion and written in standard English?

Reviewer #1: Yes

5. Review Comments to the Author

Reviewer #1: The paper “TextFormats: simplifying the definition and parsing of text formats in bioinformatics” by Giorgio Gonnella presents a framework that tries to unify different data formats commonly used in bioinformatics and builds on top of it to present a tool to specify the new format that might be used in the literature in the field. This tool was implemented in Nim programming language with wrappers to extend it in C/C++ and Python. The software provides three basic functionalities, encoding, decoding, and validation of generally defined bioinformatics formats. The results include three case studies - parsing complex alignment formats like SAM, parsing sequence-based formats, and finally, defining and parsing a new format.

Major concerns -

The choice of programming language is very peculiar. Nim is a language that is very rarely used in bioinformatics. With the growing popularity, flexibility, and user-friendliness of python-based tools and the availability of trusted legacy software written in C/C++/Java/Bash, it can be safely assumed that Nim as a language does not have much of a future in this domain. Even though the authors have developed APIs and wrappers to support other languages in TextFormats, I believe trying to use this tool in other programming languages for critical bioinformatics operation would prove unnecessarily tedious and time-consuming. An analysis showing the compilation and run-time of TextFormats in different languages might prove otherwise.

The authors have mentioned other general-purpose standard data formats such as GFA2; however, the result and discussion do not portray any comparative analysis with such format. Such analysis is required, along with the ad-hoc format parsing.

Although the authors have mentioned two major applications of the tool by working with both alignment and sequence formats, they have omitted the tool’s applicability in another major use case, feature formats such as GFF, GTF.

Minor concerns -

A visualization of the TFSL specification tree would clarify the specification representation clearly missing in the paper.

Line number 3, binary format > formats

Line 43, become > became

I.e., e.g. etc. should be followed by a comma(,).

Several minor grammatical mistakes

In conclusion, I believe that the software has novelty and can have a good prospect. However, as the premise of such a general-purpose framework is the user applications, the language should be one of current practice. Even then, I would recommend acceptance if the authors can provide substantial evidence regarding acceptable compilation and running time of the software in other languages through their developed APIs.

6. PLOS authors have the option to publish the peer review history of their article (what does this mean?). If published, this will include your full peer review and any attached files.

Reviewer #1: No

---

## [Author Response · Author response to Decision Letter 0]

2 Mar 2022

Please refer to the attached document with the responses to the reviewers and editor comments.

---

## [Decision Letter · Decision Letter 1]

11 May 2022

TextFormats: simplifying the definition and parsing of text formats in bioinformatics

PONE-D-21-34058R1

Dear Dr. Gonnella,

We’re pleased to inform you that your manuscript has been judged scientifically suitable for publication and will be formally accepted for publication once it meets all outstanding technical requirements.

Kind regards,

M. Sohel Rahman, Ph.D.

Academic Editor

PLOS ONE

Additional Editor Comments (optional):

Reviewers' comments:

Reviewer's Responses to Questions

**Comments to the Author**

1. If the authors have adequately addressed your comments raised in a previous round of review and you feel that this manuscript is now acceptable for publication, you may indicate that here to bypass the “Comments to the Author” section, enter your conflict of interest statement in the “Confidential to Editor” section, and submit your "Accept" recommendation.

Reviewer #1: All comments have been addressed

2. Is the manuscript technically sound, and do the data support the conclusions?

Reviewer #1: Yes

3. Has the statistical analysis been performed appropriately and rigorously? 

Reviewer #1: N/A

4. Have the authors made all data underlying the findings in their manuscript fully available?

Reviewer #1: Yes

5. Is the manuscript presented in an intelligible fashion and written in standard English?

Reviewer #1: Yes

6. Review Comments to the Author

Reviewer #1: I have read the revised manuscript of the submission, “TextFormats: simplifying the definition and parsing of text formats in bioinformatics,” by Giorgio Gonnella. The author has addressed most of my primary concerns regarding using nim and incorporating it with python. The installation has been made much easier in the current version. They have included three new case studies covering the comparative analysis with GFA2 and the Generic Feature Format. The visualization of the TFSL specification was much needed and has been added in Figure 1. The minor grammatical issues which were pointed out have been corrected, and now the language of the manuscript is better than before.

Parsing biological data has always been a challenge for biologists and bioinformaticians alike, and a general-purpose format parsing tool might be the solution to this age-long problem. I believe that the tool has novelty and might be used by scientists across the world. I recommend the publication of the paper in its current form.

7. PLOS authors have the option to publish the peer review history of their article (what does this mean?). If published, this will include your full peer review and any attached files.

Reviewer #1: No

---

## [Editor Report · Acceptance letter]

16 May 2022

PONE-D-21-34058R1 

TextFormats: simplifying the definition and parsing of text formats in bioinformatics 

Dear Dr. Gonnella:

I'm pleased to inform you that your manuscript has been deemed suitable for publication in PLOS ONE. Congratulations! Your manuscript is now with our production department. 

Kind regards, 

on behalf of

Dr. M. Sohel Rahman 

Academic Editor

PLOS ONE